# Healthcare utilization during acute medically attended episodes of respiratory syncytial virus-related lower respiratory tract infection among infants in the United States

Jason R. Gantenberg[1,2]*, Robertus van Aalst[1,3,4], David R. Diakun[5], Angela M. Bengtson[2], Brendan L. Limone[5], Christopher B. Nelson[6], David A. Savitz[2], Andrew R. Zullo[1,2,7]

1 Department of Health Services, Policy & Practice, Brown University School of Public Health, Providence, Rhode Island, United States of America, 2 Department of Epidemiology, Brown University School of Public Health, Providence, Rhode Island, United States of America, 3 New Products and Innovation Franchise, Vaccines Medical Affairs Sanofi, Lyon, France, 4 Department of Health Sciences, University Medical Center Groningen, Groningen, The Netherlands, 5 Merative, Ann Arbor, Michigan, United States of America, 6 Vaccines Medical Affairs, Sanofi, Swiftwater, Pennsylvania, United States of America, 7 Providence VA Medical Center, Providence, Rhode Island, United States of America

* jason_gantenberg@brown.edu

## Abstract

### Background

Respiratory syncytial virus (RSV) is the leading cause of infant hospitalization in the United States. Understanding healthcare utilization associated with medically attended (MA) RSV lower respiratory tract infection (LRTI) might inform research priorities aimed at reducing RSV-associated pediatric morbidity. We described healthcare utilization during acute MA RSV LRTI episodes within a geographically diverse cohort of infants in the United States.

### Methods

We created retrospective cohorts of infants born in the United States from July 1, 2016 through February 29, 2020 in each of three de-identified insurance claims datasets: Merative MarketScan Commercial Claims and Encounters, Multi-State MarketScan Medicaid, and Optum's de-identified Clinformatics ® Data Mart. We identified infants' first MA RSV LRTI diagnosis during their first RSV season and followed them for 7 subsequent days to record outpatient, emergency department, and inpatient hospital utilization. We calculated the number of outpatient visits, emergency department visits, and inpatient hospital stays occurring during this acute episode and estimated the proportion of episodes involving $\geq 2$ visits to a given healthcare setting.

### Results

In the CCAE database, we identified 25,409 acute MA RSV LRTI episodes under the specific RSV definition and 69,068 under the sensitive definition. In the MDCD database, these

**Data Availability Statement:** The author's funding agreements with Sanofi and AstraZeneca did not impose additional restrictions on data sharing beyond those of the data owners. Insurance claims data cannot be shared because these data are owned by third parties (Optum and Merative) and governed by their usage restrictions. Detailed information about the insurance claims data sets appear on the Merative and Optum websites; https://www.merative.com/documents/brief/marketscan-explainer-general and https://business.optum.com/en/data-analytics/life-sciences/real-world-data/claims-data.html. To purchase access to the MarketScan Commercial Claims and Encounters or MarketScan Multi-State Medicaid data sets used in this paper, researchers should contact Merative (marketscan.support@merative.com). To purchase access to Optum's de-identified Clinformatics Data Mart, researchers should contact Optum (info@optum.com). The authors received no special privileges when accessing Optum data. Employees of Merative conducted the MarketScan analyses on the same data available to customers. The surveillance data on respiratory syncytial virus we used to determine RSV season start and end dates by geographic region can be requested from the Centers for Disease Control and Prevention at https://data.cdc.gov/Laboratory-Surveillance/Percent-Positivity-of-Respiratory-Syncytial-Virus-/3cxc-4k8q/about_data). The authors have described the method to calculate these dates in a prior publication referenced in the Methods. S5 Code contains code and select (public) data used to generate the CDM analytic dataset and results (code for the Merative analyses cannot be shared due to their licensing agreements with SAS), as well as the scripts and summary data files that produce the paper, including tables and figures.

**Funding:** This work was supported by Sanofi and AstraZeneca. Employees of Sanofi (RvA, CBN) contributed to the study design, interpretation of results, and manuscript review. Sanofi also provided Brown researchers access to the Optum Clinformatics Data Mart. Both Sanofi and AstraZeneca conducted intellectual property compliance review prior to submission. The decision to submit for publication rested solely with the Brown research team.

**Competing interests:** JRG, ARZ, AMB and DAS receive research funding support from Sanofi (awarded to and administered by Brown University). DRD and BLL are employees of Merative, who was contracted by Sanofi to perform analyses included in this paper. CBN and RvA are

totals were 67,357 and 170,744, while in the CDM database, they were 12,402 and 31,363, respectively. Across data sources, 34%–69% of infants' first acute MA RSV LRTI episodes involve 2 or more visits to a healthcare setting within 7 days. The percentage of episodes involving at least 2 visits ranged from 34–62% among healthy term infants, 38–65% for Palivizumab-eligible infants, and 38–69% for infants with other comorbidities.

## Conclusions

Within a week of their first MA RSV LRTI diagnosis, infants frequently experience at least 2 visits to one or more healthcare settings, regardless of their comorbidity profile. The percentage of MA RSV LRTI episodes involving at least 2 visits to a healthcare setting may vary by insurance claims database, even between commercial payers.

## Introduction

Respiratory syncytial virus (RSV) is a major contributor to childhood morbidity worldwide [1]. Among young children in the United States, RSV drives seasonal respiratory illness-related hospitalization rates [2] and remains the leading cause of hospitalization among infants in the United States [3, 4].

The risk of severe RSV outcomes is inversely associated with chronological age, peaking in the first few months of life and progressively decreasing as the child ages [5–7]. Preterm infants and those with comorbidities such as chronic lung disease of prematurity (CLD) or hemodynamically significant congenital heart disease (HS-CHD) are at higher risk of severe complications due to RSV infection [8]. Consequently, the scientific literature focuses primarily on RSV-related outcomes among high-risk infants [9]. Until August 2023, when the Advisory Committee on Immunization Practices recommended monoclonal antibody administration for most infants entering their first RSV season and infants and young children at higher risk entering their second [10], pre-exposure prophylaxis (palivizumab) was recommended only for infants born at gestational age <29 weeks and some infants with CLD or HS-CHD [7].

Recent data indicate that healthy full-term infants account for over 70% of medically attended (MA) RSV episodes during infants' first RSV season [7, 11]. However, studies of RSV-related healthcare utilization and burden frequently omit term infants without comorbidities that would put them at higher risk of severe outcomes [9]. To quantify the healthcare burden of RSV among all infants, we must include those at lower risk of severe infection but who still account for the majority of MA RSV lower respiratory tract infection (LRTI) cases.

Given the large public health burden of RSV, characterizing the associated healthcare utilization is an important aspect of understanding how this burden manifests among infants with MA illness. This information may guide the design of prevention efforts, including those associated with monoclonal antibodies and vaccines [12].

In this study, we described healthcare utilization during infants' first acute MA RSV LRTI episodes during their first RSV season, focusing on the places of service utilized during these episodes. We evaluated 1) the number of visits to each place of service during acute MA RSV LRTI episodes; 2) the proportion of acute MA RSV LRTI episodes involving at least one visit to a given place of service; and 3) the proportion of MA RSV LRTI episodes involving ≥ 2 visits to a healthcare setting. We compared these results across three insurance claims databases

employees of Sanofi and may hold shares and/or stock options in the company.

**Abbreviations:** Abbreviation, Definition; MA, medically attended; RSV, respiratory syncytial virus; LRTI, lower respiratory tract infection; CCAE, Merative MarketScan ® Commercial Claims and Encounters; MDCD, Multi-State MarketScan ® Medicaid; CDM, Optum's de-identified Clinformatics ® Data Mart; ED, emergency department; CLD, chronic lung disease of prematurity; HS-CHD, hemodynamically significant congenital heart disease.

capturing different geographic and socioeconomic subsets of infants in the United States and stratified estimates by infants' comorbidity profiles.

## Methods

### Study population

We conducted a retrospective cohort study of infants born in the United States between July 1, 2016 and February 29, 2020. We identified live births occurring within this time window and constructed retrospective birth cohorts in each of three de-identified insurance claims datasets —Merative MarketScan Commercial Claims and Encounters ® (CCAE), Multi-State Market-Scan Medicaid ® (MDCD), and Optum's de-identified Clinformatics ® Data Mart (CDM). These datasets capture different geographic distributions of infants and implicitly capture different socioeconomic strata (commercial CCAE/CDM datasets versus Medicaid). We included infants in the current analysis if they had a qualifying MA RSV LRTI diagnosis during their first RSV season. We defined their first such diagnosis as the "index diagnosis", which triggered the start of their first acute MA RSV LRTI episode. We followed infants for 7 days after this index diagnosis to evaluate their healthcare utilization, assuming no loss to follow-up during this short time window.

### Comorbidity groups

We classified infants into one of three mutually exclusive comorbidity groups based on gestational age and comorbidity status, as described in prior work [11]. Briefly, infants were categorized as follows:

- *A. Healthy term infants*: infants with no indication of having been delivered preterm, based either on diagnosis related group (DRG) codes identifying a full-term neonate without major problems or the absence of any DRG or International Classification of Diseases-Tenth Revision-Clinical Modification (ICD-10-CM) codes indicating preterm birth or gestational age less than 37 weeks. We categorized infants without any such DRG or ICD-10-CM codes (*i.e.*, infants with unknown gestational age) to be full term, as infants born preterm would likely have an accompanying code.

- *B. Palivizumab-eligible*: infants who would be eligible to receive palivizumab, either because they had a gestational age less than 29 weeks, a qualifying combination of preterm birth and CLD, or, HS-CHD.

- *C. Other comorbidities*: infants of any gestational age with comorbidities (*e.g.*, lower-risk congenital heart disease, respiratory conditions) that might predispose them to severe RSV-related outcomes.

For comorbidity groups B and C, we included infants with relevant ICD-10-CM codes if that code appeared at any time prior to the onset of their first MA RSV LRTI episode. **S1 and S2 Tables in S1 File** list the diagnostic codes underlying these categories.

### Acute RSV episodes

As in our prior work [11], we identified each infants' first RSV season by calculating beginning and end dates specific to the Census division in which the infant was born, according to the "retrospective slope 10" procedure described by Midgley et al. [13, 14]. Briefly, this method calculates a normalized 5-week moving average of RSV detections and assigns the season's start date to the second of two consecutive weeks in which the increase in the moving average

exceeds 10. The season offset date is then set as the last week in which the 5-week moving average remains above that during the week of season onset.

We defined an "acute MA RSV LRTI episode" as the 0–7 days (inclusive) following the occurrence of an index MA RSV LRTI diagnosis, where day 0 is the date of this index diagnosis. Hereafter we refer to the first acute MA RSV LRTI episode during an infants' first RSV season as the "first MA RSV LRTI episode". We focused on infants' first MA RSV LRTI episode because our preliminary analyses preparatory to research suggested that second MA episodes within the same RSV season were rare.

Also similar to our prior work [11], we used two alternative definitions of the index MA RSV LRTI diagnosis to reflect uncertainty in the ascertainment of MA RSV LRTI stemming from a lack of systematic laboratory data on RSV infection in the insurance claims databases. Both definitions were based on combinations of ICD-10-CM diagnosis codes (**S3 Table in S1 File**):

- The *specific* definition included ICD-10-CM codes that explicitly mention RSV (B974, J121, J205, and J210), where a diagnosis of B974 ("RSV as the cause of diseases classified elsewhere") attached to an outpatient or emergency department (ED) visit qualified only if the infant had another respiratory condition diagnosed within ± 5 days.

- The *sensitive* definition included all codes from the *specific* definition, plus 2 additional codes capturing unspecified bronchiolitis (J218 and J219). The CCAE database includes the first 4 diagnosis positions for outpatient and ED visits and the first 15 diagnosis positions for inpatient hospital stays; we limited diagnosis ascertainment to these positions in the CDM and MDCD data to harmonize analyses across the three databases.

For each acute MA RSV LRTI episode, we identified outpatient visits, ED visits, and inpatient hospital stays attached to a sensitive RSV diagnosis occurring during the episode (regardless of the index diagnosis definition). To reduce the impact of coding errors in claims and to avoid overestimating healthcare utilization, we allowed for a maximum of one outpatient and one ED visit per day; therefore, infants could have up to 8 outpatient and/or ED visits recorded during the episode, inclusive of the index date. We counted inpatient stays if they overlapped with any part of the 7-day episode window. If an inpatient stay began on the same day that an outpatient or ED visit occurred, we counted each event as a separate instance of healthcare utilization in order to characterize the different settings infants contact during the acute RSV episode. In the CDM database, we identified inpatient stays using a confinement identification code available in the base dataset. For a subset of inpatient events without associated confinement codes, we chained events with contiguous dates of service together into a single inpatient stay. In all databases, a new inpatient stay was triggered only when we found a gap of at least one day between the discharge date of one stay and the admission date for the next. **S4 Table in S1 File** lists the place of service codes used to classify events as outpatient, ED, or inpatient.

## Descriptive analysis

We summarized the number of healthcare contacts at each place of service by summing individual outpatient, ED, and inpatient events and calculating the mean number of visits/stays across acute MA RSV LRTI episodes. We calculated the proportion of episodes involving 0, 1, $\geq 1$, and/or $\geq 2$ contacts with a given place of service and quantified the uncertainty due to random error around these estimated proportions using approximate 95% confidence intervals [15]. In addition to estimating marginal quantities within each claims database, we stratified estimates by comorbidity group.

### Ethics approval

This study involved only secondary analysis of de-identified insurance claims data and is not considered human subjects research for that reason. No institutional review board approval was required.

## Results

### Descriptive characteristics

In the CCAE database, we identified 25,409 acute MA RSV LRTI episodes under the specific RSV definition and 69,068 under the sensitive definition. In the MDCD database, we identified 67,357 and 170,744 such episodes, respectively. In the CDM database, we identified 12,402 and 31,363 such episodes. Geographic distributions and insurance plan types differed between the commercial databases. Across data sources and RSV definitions, approximately 3% of infants had HS-CHD, < 1% had CLD, and 9–13% had a chronic condition of any kind. See **Table 1** for demographic and clinical characteristics among infants with acute RSV episodes under each definition.

### Participants

In the final CCAE analytic sample 76.7% of infants were classified as comorbidity group A (healthy term), 3.8% comorbidity group B (palivizumab-eligible), and 19.4% comorbidity group C (other comorbidities). In the final MDCD analytic sample, these percentages were 77.7%, 3.9%, and 18.3%, respectively, while in the CDM data they were 79.4%, 3.3%, and 17.2%.

### Healthcare utilization during acute RSV episodes

**Specific definition.**   Under the specific definition, infants averaged 1.11, 0.9, and 1.57 outpatient visits in the CCAE, MDCD, and CDM databases, respectively. They averaged 0.36, 0.52, and 0.4 ED visits, and 0.24, 0.22, 0.24 inpatient stays per acute RSV episode. Eighty percent, 68%, and 94% of episodes involved at least one outpatient visit in the CCAE, MDCD and CDM databases, respectively. Thirty-three percent, 45%, and 36% had at least one ED visit, while 22%, 20%, and 24% had at least one inpatient stay (**Fig 1**). Across the databases, 43–63% of episodes involved ≥ 2 visits to any healthcare setting, varying by database and comorbidity group (**Fig 2**).

**Sensitive definition.**   Under the sensitive definition, infants averaged 1.19, 0.89, and 1.4 outpatient visits in the CCAE, MDCD, and CDM databases, respectively. They averaged 0.24, 0.48, and 0.24 ED visits, and 0.12, 0.13, 0.12 inpatient stays per acute RSV episode. Eighty-nine percent, 69%, and 96% of episodes involved at least one outpatient visit in the CCAE, MDCD and CDM databases, respectively. Twenty-two percent, 43%, and 22% had at least one ED visit, while 11%, 12%, and 12% had at least one inpatient stay (**Fig 1**). Across the databases, 35–44% of episodes involved ≥ 2 visits to any healthcare setting, varying by database and comorbidity group (**Fig 2**).

Compared to the other databases, infants in the MDCD database had lower usage of the outpatient setting but higher usage of the ED and, in some cases, the inpatient setting (**Fig 1**).

### Healthcare utilization by comorbidity group

Healthy term infants (comorbidity group A) accounted for the largest number of visits to all healthcare settings during acute MA RSV LRTI episodes, followed by infants with other

**Table 1. Characteristics of infants with an acute MA RSV LRTI during their first RSV season, by insurance claims database and index diagnosis definition.**

| Variable | MarketScan Commercial | | | | MarketScan Medicaid | | | | Optum Clinformatics | | | |
|---|---|---|---|---|---|---|---|---|---|---|---|---|
| | Specific | | Sensitive | | Specific | | Sensitive | | Specific | | Sensitive | |
| | N | % | N | % | N | % | N | % | N | % | N | % |
| *Birth month* | | | | | | | | | | | | |
| January | 1,019 | 4.0 | 2,576 | 3.7 | 2,651 | 4.0 | 6,202 | 3.9 | 537 | 4.3 | 1,073 | 3.4 |
| February | 459 | 1.8 | 1,218 | 1.8 | 1,005 | 1.8 | 2,697 | 1.7 | 188 | 1.5 | 448 | 1.4 |
| March | 199 | 0.8 | 720 | 1.0 | 394 | 0.8 | 1,174 | 0.7 | 65 | 0.5 | 159 | 0.5 |
| April | 782 | 3.1 | 2,239 | 3.2 | 1,706 | 3.1 | 4,840 | 3.0 | 249 | 2.0 | 698 | 2.2 |
| May | 2,457 | 9.7 | 7,356 | 10.7 | 5,431 | 9.7 | 15,235 | 9.4 | 1,078 | 8.7 | 3,017 | 9.6 |
| June | 2,641 | 10.4 | 7,740 | 11.2 | 5,850 | 10.4 | 16,055 | 9.9 | 1,202 | 9.7 | 3,419 | 10.9 |
| July | 3,634 | 14.3 | 10,890 | 15.8 | 9,019 | 14.3 | 24,386 | 15.1 | 1,694 | 13.7 | 4,816 | 15.4 |
| August | 3,750 | 14.8 | 10,527 | 15.2 | 9,825 | 14.8 | 24,853 | 15.4 | 1,764 | 14.2 | 4,825 | 15.4 |
| September | 3,489 | 13.7 | 9,120 | 13.2 | 9,382 | 13.7 | 22,028 | 13.7 | 1,653 | 13.3 | 4,285 | 13.7 |
| October | 3,100 | 12.2 | 7,733 | 11.2 | 8,898 | 12.2 | 19,308 | 12.0 | 1,698 | 13.7 | 3,989 | 12.7 |
| November | 2,433 | 9.6 | 5,593 | 8.1 | 7,098 | 9.6 | 15,021 | 9.3 | 1,355 | 10.9 | 2,832 | 9.0 |
| December | 1,446 | 5.7 | 3,356 | 4.9 | 4,411 | 5.7 | 9,479 | 5.9 | 919 | 7.4 | 1,802 | 5.8 |
| *Birth year* | | | | | | | | | | | | |
| 2016 | 4,510 | 17.8 | 12,990 | 18.8 | 13,409 | 20.4 | 33,709 | 20.9 | 2,182 | 17.6 | 5,742 | 18.3 |
| 2017 | 6,683 | 26.3 | 19,739 | 28.6 | 17,611 | 26.8 | 45,884 | 28.4 | 3,315 | 26.7 | 8,865 | 28.3 |
| 2018 | 7,002 | 27.6 | 19,400 | 28.1 | 16,243 | 24.7 | 40,216 | 24.9 | 3,401 | 27.4 | 8,833 | 28.2 |
| 2019 | 7,125 | 28.0 | 16,755 | 24.3 | 17,855 | 27.2 | 40,289 | 25.0 | 3,433 | 27.7 | 7,802 | 24.9 |
| 2020 | 89 | 0.3 | 184 | 0.3 | 552 | 0.8 | 1,180 | 0.7 | 71 | 0.6 | 121 | 0.4 |
| *Sex* | | | | | | | | | | | | |
| Female | 11,321 | 44.6 | 29,270 | 42.4 | 29,810 | 54.6 | 70,886 | 44.0 | 5,484 | 44.2 | 13,292 | 42.4 |
| Male | 14,088 | 55.4 | 39,798 | 57.6 | 35,852 | 45.4 | 90,373 | 56.0 | 6,916 | 55.8 | 18,065 | 57.6 |
| Unknown | 0 | 0.0 | 0 | 0.0 | 8 | 0.0 | 19 | 0.0 | 2 | 0.0 | 6 | 0.0 |
| *Census division[a]* | | | | | | | | | | | | |
| East North Central | 3,994 | 15.7 | 12,100 | 17.5 | - | - | - | - | 1,593 | 12.8 | 4,635 | 14.8 |
| East South Central | 2,144 | 8.4 | 4,577 | 6.6 | - | - | - | - | 546 | 4.4 | 1,159 | 3.7 |
| Mid Atlantic | 3,206 | 12.6 | 10,642 | 15.4 | - | - | - | - | 799 | 6.4 | 2,379 | 7.6 |
| Mountain | 1,492 | 5.9 | 4,390 | 6.4 | - | - | - | - | 1,091 | 8.8 | 3,277 | 10.4 |
| New England | 812 | 3.2 | 2,616 | 3.8 | - | - | - | - | 299 | 2.4 | 877 | 2.8 |
| Other/Unknown | 171 | 0.7 | 411 | 0.6 | - | - | - | - | 39 | 0.3 | 134 | 0.4 |
| Pacific | 930 | 3.7 | 3,232 | 4.7 | - | - | - | - | 631 | 5.1 | 1,983 | 6.3 |
| South Atlantic | 5,968 | 23.5 | 14,778 | 21.4 | - | - | - | - | 2,457 | 19.8 | 6,065 | 19.3 |
| West North Central | 2,191 | 8.6 | 5,872 | 8.5 | - | - | - | - | 1,932 | 15.6 | 4,529 | 14.4 |
| West South Central | 4,501 | 17.7 | 10,450 | 15.1 | - | - | - | - | 3,015 | 24.3 | 6,325 | 20.2 |
| *Comorbidity group* | | | | | | | | | | | | |
| A: Healthy term | 19,496 | 76.7 | 52,947 | 76.7 | 51,064 | 77.8 | 124,788 | 77.4 | 9,850 | 79.4 | 25,040 | 79.8 |
| B: Palivizumab-eligible | 978 | 3.9 | 2,906 | 4.2 | 2,565 | 3.9 | 7,227 | 4.5 | 414 | 3.3 | 1,053 | 3.4 |
| C: Other comorbidities | 4,935 | 19.4 | 13,215 | 19.1 | 12,041 | 18.3 | 29,263 | 18.1 | 2,138 | 17.2 | 5,270 | 16.8 |
| *Gestational age* | | | | | | | | | | | | |
| < 29 weeks | 171 | 0.7 | 614 | 0.9 | 724 | 1.1 | 2,248 | 1.4 | 7 | 0.1 | 19 | 0.1 |
| 29–31 weeks | 307 | 1.2 | 742 | 1.1 | 1,075 | 1.6 | 2,504 | 1.6 | 37 | 0.3 | 68 | 0.2 |
| 32–36 weeks | 2,278 | 10.9 | 6,872 | 9.9 | 7,956 | 12.1 | 18,503 | 11.5 | 945 | 7.6 | 2,295 | 7.3 |
| Preterm, unknown GA | 580 | 2.3 | 1,560 | 2.3 | 1,291 | 2.0 | 3,146 | 2.0 | 562 | 4.5 | 1,346 | 4.3 |
| Full term, >36 weeks | 15,091 | 59.4 | 40,832 | 59.1 | 37,363 | 56.9 | 91,485 | 56.7 | 7,909 | 63.8 | 20,112 | 64.1 |
| Unknown GA | 6,482 | 25.5 | 18,448 | 26.7 | 17,261 | 26.3 | 43,392 | 26.9 | 2,942 | 23.7 | 7,523 | 24.0 |

*(Continued)*

**Table 1.** (Continued)

| Variable | MarketScan Commercial | | | | MarketScan Medicaid | | | | Optum Clinformatics | | | |
|---|---|---|---|---|---|---|---|---|---|---|---|---|
| | Specific | | Sensitive | | Specific | | Sensitive | | Specific | | Sensitive | |
| | N | % | N | % | N | % | N | % | N | % | N | % |
| *Plan type* | | | | | | | | | | | | |
| EPO/PPO | 13,742 | 54.1 | 36,644 | 53.0 | 42 | 0.1 | 85 | 0.0 | 1,318 | 10.6 | 3,241 | 10.3 |
| CDHP/HDHP | 5,533 | 21.8 | 15,256 | 22.1 | 0 | 0.0 | 0 | 0.0 | 3,619 | 29.2 | 9,442 | 30.1 |
| HMO | 2,500 | 9.8 | 7,335 | 10.6 | 40,769 | 62.1 | 101,338 | 62.8 | 1,053 | 8.5 | 2,517 | 8.0 |
| POS/POS with capitation | 2,691 | 10.6 | 7,347 | 10.6 | 0 | 0.0 | 0 | 0.0 | 6,288 | 50.7 | 15,869 | 50.6 |
| Comprehensive/indemnity | 411 | 1.6 | 1,051 | 1.5 | 24,744 | 37.7 | 59,588 | 37.0 | 1 | 0.0 | 1 | 0.0 |
| Missing/Unknown | 532 | 2.1 | 1,435 | 2.1 | 115 | 0.2 | 267 | 0.2 | 123 | 1.0 | 293 | 0.9 |
| *HS-CHD* | 794 | 3.1 | 2,293 | 3.3 | 1,840 | 2.8 | 5,087 | 3.1 | 343 | 2.8 | 888 | 2.8 |
| *Chronic lung disease* | 168 | 0.7 | 652 | 0.9 | 563 | 0.9 | 1,967 | 1.2 | 68 | 0.6 | 181 | 0.6 |
| *Any chronic condition[b]* | 3,039 | 12.0 | 9,211 | 13.3 | 5,972 | 9.1 | 16,682 | 10.5 | 1,392 | 11.2 | 3,529 | 11.2 |

Abbreviations: CDHP, consumer-driven health plan; CHD, congenital heart disease; EPO, exclusive provider organization; GA, gestational age; HDHP, high-deductible health plan; HMO, health maintenance organization; LTFU, lost to follow-up; MA RSV LRTI, medically attended respiratory syncytial virus lower respiratory tract infection; POS, point of service; PPO, preferred provider organization.

[a] Census division unavailable in MarketScan Medicaid database.

[b] Includes chronic lung disease and hemodynamically significant congenital heart disease.

comorbidities (comorbidity group C) and palivizumab-eligible infants (comorbidity group B) (Table 2, Fig 1).

Healthy term infants had the highest number of outpatient visits across claims databases and RSV definitions and the highest proportion of MA RSV LRTI episodes involving the outpatient setting. Infants with other comorbidities had comparable outpatient usage in most cases. Palivizumab-eligible infants consistently exhibited the highest proportion of MA RSV LRTI episodes involving an inpatient stay, followed by infants with other comorbidities and then healthy term infants.

Under the specific index diagnosis definition, infants did not appear to differ meaningfully by comorbidity group in their usage of the ED. Under the sensitive definition in commercial claims, palivizumab-eligible infants and those with other comorbidities exhibited higher ED utilization than healthy term infants. The magnitudes of the differences in healthcare utilization between comorbidity groups varied by claims database, although less so for ED visits.

Infants with other comorbidities were the most likely to have ≥ 2 visits to healthcare settings during their first MA RSV LRTI episode, followed by palivizumab-eligible infants and healthy term infants, respectively. Regardless of comorbidity group, infants' RSV episodes frequently involved multiple visits to healthcare settings (Table 2, Fig 2).

## Discussion

Between one- and two-thirds of infants in our study had ≥ 2 visits to healthcare settings during their first MA RSV LRTI episode. The majority of episodes involved at least one visit to an outpatient setting, while a substantial proportion involved visits to the ED and/or inpatient hospital stays. Up to 1 in 5 infants were hospitalized, including 9–21% of healthy term infants without known comorbidities, as reported in our prior work [11]. While infants who were eligible for palivizumab or had other comorbidities were more likely to be hospitalized, the finding that between 1 in 5–10 healthy term infants experience an inpatient stay during their MA RSV LRTI episode suggests substantial healthcare utilization across comorbidity profiles,

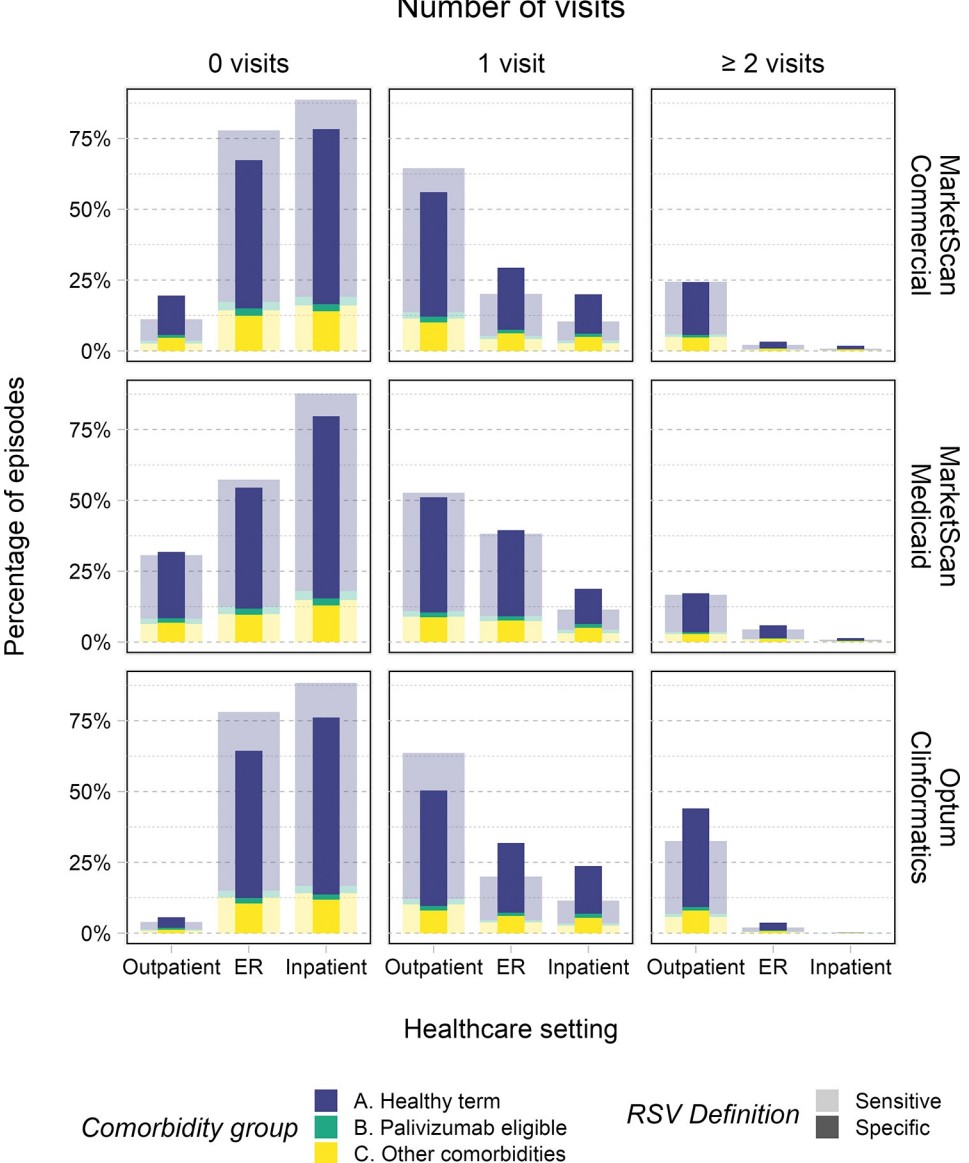

**Fig 1. Percentage of acute MA RSV LRTI episodes involving 0, 1, or ≥ 2 visits to a given healthcare setting, stratified by RSV index diagnosis definition and insurance claims database.** Fill colors depict the share of each bar accounted for by a given comorbidity group.

consistent with prior work [5, 11, 16]. Given their large numbers, healthy term infants drive much of the related clinical and public health burden of RSV.

Because RSV is the leading cause of infant hospitalization in the United States [4], healthcare utilization associated with infant MA RSV LRTI episodes outpaces that of many other diseases, either per episode or due simply to the volume of RSV episodes that occur in a typical year. Prior research in a small cohort (*n* = 360) suggests that infants hospitalized for respiratory illness more frequently test positive for RSV than infants in ED or outpatient settings [17, 18]. These studies, too, found that most RSV episodes involved more than one visit to a healthcare setting. Future prevention policies should weigh how infants with different comorbidity profiles contribute to overall healthcare utilization and cost, under explicitly defined optimization

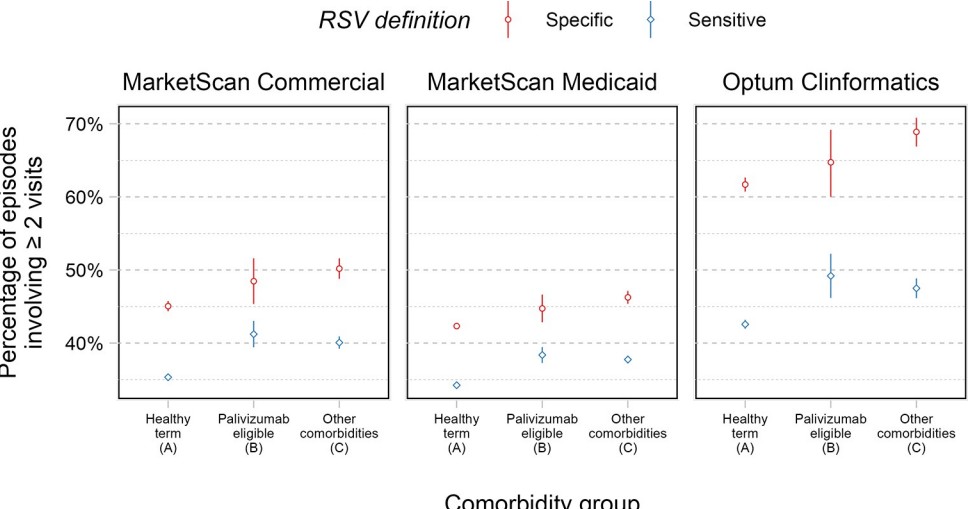

**Fig 2. Overall percentage of acute MA RSV LRTI episodes involving ≥ 2 visits to any healthcare setting(s), stratified by comorbidity group and insurance claims database.** Line ranges depict 95% confidence intervals.

goals. Policies aimed at reducing overall RSV-associated healthcare utilization and costs—for instance, via monoclonal antibodies recently approved by the FDA [19]—should consider that healthy term infants and those with non-palivizumab-eligible comorbidities comprise a majority of the infants who account for this utilization. Such a policy would focus on reducing the overall burden of MA RSV LRTI, in contrast to risk-based policies aimed at avoiding the worst outcomes among subsets of infants at especially high risk of severe infection.

We also found that healthcare utilization patterns during MA RSV LRTI episodes differed by payer (**Table 2**), a finding that aligns with our previous analysis [11] and prior literature [18]. Infants enrolled in Medicaid were more likely to have an episode involving an ED visit and less likely to have one involving an outpatient visit, compared to infants in commercial insurance claims; Medicaid infants are more likely to use the ED as their initial point of contact with the healthcare system (versus, say, a visit to a primary care provider) [18]. Comparing commercial insurance claims, infants in the Optum Clinformatics database averaged a larger proportion of episodes involving ≥ 2 visits to a healthcare setting compared to infants in MarketScan Commercial (**Fig 2**). Whether these differences derive from administrative differences in the databases, varying infant populations, or, perhaps, imperfect harmonization of our data analyses across databases, is difficult to say. Future comparative analyses of claims and surveillance databases may be informative. A recent systematic literature review on RSV hospitalization rates among infants and young children showed that estimates appear to vary by data source and by RSV ascertainment method [20], suggesting that some residual uncertainty regarding RSV's burden on the healthcare system could be reduced by improving the quality and comprehensiveness of RSV diagnosis data. Furthermore, there is a paucity of data on outpatient and ED use related to RSV infections [9], and our study describes how frequently these settings are utilized during infants' first MA RSV LRTI episodes.

## Limitations

The following limitations should be kept in mind when interpreting the results of our study. First, we focus only on acute MA RSV LRTI episodes, which we defined as the 0–7 days following the index diagnosis. However, healthcare utilization associated with the onset of infection

**Table 2. Percentage of MA RSV LRTI episodes that involved 0, 1, or 265 2 visits to a given healthcare setting, stratified by insurance claims database, index diagnosis definition, and comorbidity group.**

| Comorbidity group | Setting | MarketScan Commercial (CCAE) | | | | | MarketScan Medicaid (MDCD) | | | | | Optum Clinformatics (CDM) | | | | |
|---|---|---|---|---|---|---|---|---|---|---|---|---|---|---|---|---|
| | | Episodes (#) | Visits (#) | 0 | 1 | ≥ 2 | Episodes (#) | Visits (#) | 0 | 1 | ≥ 2 | Episodes (#) | Visits (#) | 0 | 1 | ≥ 2 |
| | | | | Row (%) | | | | | Row (%) | | | | | Row (%) | | |
| **Index Diagnosis: Specific Definition** | | | | | | | | | | | | | | | | |
| A. Healthy term | Outpatient | 19,496 | 21,865 | 18.2 | 57.4 | 24.4 | 52,348 | 48,199 | 30.0 | 52.3 | 17.7 | 9,850 | 15,472 | 4.8 | 51.4 | 43.8 |
| | Emergency dept. | | 6,829 | 68.3 | 28.5 | 3.1 | | 27,011 | 55.0 | 39.2 | 5.9 | | 3,776 | 65.5 | 31.0 | 3.5 |
| | Inpatient | | 4,109 | 80.5 | 18.0 | 1.5 | | 9,684 | 82.7 | 16.1 | 1.2 | | 2,124 | 78.6 | 21.3 | 0.1 |
| B. Palivizumab eligible | Outpatient | 978 | 992 | 25.9 | 51.5 | 22.6 | 2,649 | 2,015 | 41.3 | 44.2 | 14.5 | 414 | 560 | 17.4 | 46.6 | 36.0 |
| | Emergency dept. | | 379 | 65.0 | 31.6 | 3.4 | | 1,311 | 56.2 | 38.7 | 5.2 | | 194 | 59.4 | 34.8 | 5.8 |
| | Inpatient | | 345 | 67.1 | 30.6 | 2.4 | | 1,057 | 63.0 | 34.1 | 2.8 | | 181 | 57.0 | 42.3 | 0.7 |
| C. Other comorbidities | Outpatient | 4,935 | 5,295 | 23.7 | 51.8 | 24.5 | 12,360 | 10,201 | 37.1 | 47.5 | 15.5 | 2,138 | 3,400 | 7.3 | 46.4 | 46.4 |
| | Emergency dept. | | 1,962 | 64.2 | 32.1 | 3.7 | | 6,784 | 52.3 | 41.3 | 6.5 | | 946 | 60.5 | 35.1 | 4.4 |
| | Inpatient | | 1,525 | 71.8 | 25.6 | 2.6 | | 3,905 | 70.7 | 27.1 | 2.2 | | 679 | 68.4 | 31.4 | 0.2 |
| **Index Diagnosis: Sensitive Definition** | | | | | | | | | | | | | | | | |
| A. Healthy term | Outpatient | 53,099 | 63,580 | 9.8 | 66.2 | 24.0 | 132,070 | 120,933 | 28.9 | 54.1 | 17.0 | 25,040 | 35,119 | 3.3 | 64.5 | 32.1 |
| | Emergency dept. | | 12,400 | 78.8 | 19.2 | 2.0 | | 61,831 | 58.0 | 37.6 | 4.4 | | 5,782 | 78.9 | 19.2 | 1.9 |
| | Inpatient | | 5,354 | 90.6 | 8.7 | 0.7 | | 13,797 | 90.2 | 9.1 | 0.7 | | 2,586 | 89.8 | 10.2 | 0.1 |
| B. Palivizumab eligible | Outpatient | 2,853 | 2,998 | 23.9 | 53.0 | 23.1 | 7,692 | 5,613 | 43.5 | 42.9 | 13.6 | 1,053 | 1,362 | 12.1 | 57.5 | 30.5 |
| | Emergency dept. | | 865 | 72.1 | 25.7 | 2.2 | | 3,758 | 55.6 | 40.3 | 4.1 | | 330 | 71.9 | 25.1 | 3.0 |
| | Inpatient | | 800 | 73.5 | 24.9 | 1.6 | | 2,501 | 69.2 | 29.1 | 1.7 | | 260 | 75.8 | 23.7 | 0.5 |
| C. Other comorbidities | Outpatient | 13,116 | 15,549 | 13.9 | 60.2 | 25.9 | 30,982 | 26,269 | 34.8 | 49.2 | 16.0 | 5,270 | 7,500 | 5.2 | 60.3 | 34.5 |
| | Emergency dept. | | 3,593 | 75.3 | 22.2 | 2.5 | | 15,754 | 54.6 | 40.4 | 5.0 | | 1,469 | 75.0 | 22.4 | 2.6 |
| | Inpatient | | 2,199 | 84.4 | 14.4 | 1.2 | | 6,039 | 81.8 | 16.9 | 1.3 | | 842 | 84.1 | 15.7 | 0.1 |

Abbreviations: MA, medically attended; RSV, respiratory syncytial virus; LRTI lower respiratory tract infection

may extend beyond this initial episode. A study by Doucette et al. found that up to one-third of infants hospitalized for RSV are readmitted to the hospital within 3 months [21].

Second, insurance claims databases generally do not contain information on laboratory testing for RSV infection, and so we used diagnostic codes as proxies. Misclassification of RSV status varies by healthcare setting, with inpatient diagnoses being most accurate [22]. We addressed the inherent limitations of code-based proxies by implementing both a specific and a sensitive definition for the index RSV diagnosis, "bounding" our estimates under extreme assumptions regarding the accuracy of ICD-10-CM codes. Given that a substantial proportion of non-RSV-coded bronchiolitis diagnoses are probably RSV-related [23], our specific definition likely underestimates the true RSV burden, while our sensitive definition assumes all unspecified bronchiolitis diagnoses are RSV-related. Under the sensitive definition, episodes were less likely to involve an ED visit or inpatient hospital stay, suggesting this definition captures milder MA respiratory episodes.

Third, our definition of palivizumab eligibility only approximates current eligibility guidelines [7], due again to the limitations of insurance claims data. Infants' gestational age and/or comorbidity status may have been misclassified, possibly understating differences in healthcare utilization patterns by comorbidity group. Similarly, we assume that infants with

unknown gestational age (24–27%) indicate a full-term birth. The elicited proportions of full-term births across insurance claims databases ranged from 83.6% (MDCD) to 88% (CDM), in good agreement with recent data from the National Center for Healthcare Statistics (just under 90%) [24], though we would not expect Medicaid infants to be representative of the broader population.

Fourth, we did not account for potential selection bias or loss to follow-up [25, 26]. Selection bias could have occurred if infants who lost insurance eligibility before or during their first RSV season differed from infants who were observed until the end of their first season. In our prior work, weighting for loss of insurance eligibility had modest effects on most estimates of MA RSV LRTI burden [11].

Fifth, we observed notable differences between the commercial databases (CCAE and CDM) in the proportion of acute RSV episodes without an outpatient visit and in the proportion of episodes involving $\geq 2$ visits to a healthcare setting. We speculate that these differences could be explained by variation in the geographic distribution or payer types between the data sources but cannot rule out subtle differences in how analyses were implemented across databases.

Sixth, we cannot rule out that some infants in our analytic sample appear in multiple insurance claims databases. We suspect the overlap to be negligible, particularly over the timeframe of infants' first MA RSV LRTI episode.

Seventh, our acute RSV episode definition might have missed healthcare utilization related to the RSV infection but which preceded an infant's meeting the index RSV diagnosis criteria. Healthcare utilization associated with the RSV episode may also have occurred beyond the 7-day episode window we investigated. Therefore, our estimates may prove to be slightly conservative.

Finally, the SARS-CoV-2 pandemic markedly altered RSV transmission patterns among infants [27–29]. We ended our birth cohort to February 29, 2020 to focus on healthcare utilization under typical seasonal RSV dynamics, and our estimates may not apply to healthcare utilization occurring during atypical waves of RSV infection.

## Conclusions

Infants' first acute MA RSV LRTI episodes frequently involve $\geq 2$ visits to a healthcare setting. Healthcare utilization during these episodes is not limited to infants considered to be at high risk for severe RSV-related complications under current pre-exposure prophylaxis guidelines. Prevention efforts focused on reducing the overall burden of RSV to the healthcare system should consider 1) that healthcare utilization differs by payer and 2) that healthy term infants and those with less severe comorbidities account for the majority of healthcare visits during acute MA RSV LRTI episodes.

## Supporting information

**S1 File.** (1) Classification of gestational age groups based on diagnoses attached to birth hospitalization. (2) ICD-10-CM codes assessed for presence of comorbidities. (3) Codes used to classify MA RSV LRTI using the *specific* and *sensitive* definitions. (4) Place of service classifications for MA RSV LRTI diagnoses identified in MarketScan Commercial, MarketScan Medicaid, and Optum Clinformatics.
(PDF)

**S2 File. Code used in the Optum analysis and code and summary data files underlying the paper.**
(ZIP)

## Acknowledgments

The authors thank Monika Reddy Bhuma for conducting supplemental literature review, as well as Jeffrey Miller, David M. Smith, and Rebecca A. Bachmann, for their comments on elements of the manuscript.

## Prior presentation

Portions of the data reported in this paper were presented at IDWeek 2022, October 19–23, 2022, Washington, D.C.

## Author Contributions

**Conceptualization:** Jason R. Gantenberg, Robertus van Aalst, Angela M. Bengtson, Brendan L. Limone, Christopher B. Nelson, David A. Savitz, Andrew R. Zullo.

**Formal analysis:** Jason R. Gantenberg, David R. Diakun.

**Funding acquisition:** Andrew R. Zullo.

**Methodology:** Jason R. Gantenberg, Robertus van Aalst, David R. Diakun, Angela M. Bengtson, Christopher B. Nelson, David A. Savitz, Andrew R. Zullo.

**Software:** Jason R. Gantenberg, David R. Diakun.

**Supervision:** Robertus van Aalst, Andrew R. Zullo.

**Validation:** Jason R. Gantenberg, David R. Diakun.

**Visualization:** Jason R. Gantenberg.

**Writing – original draft:** Jason R. Gantenberg.

**Writing – review & editing:** Jason R. Gantenberg, Robertus van Aalst, David R. Diakun, Angela M. Bengtson, Brendan L. Limone, Christopher B. Nelson, David A. Savitz, Andrew R. Zullo.

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
