## [Decision Letter · Decision Letter 0]

7 Jul 2024

PONE-D-24-01606Healthcare Utilization During Acute Medically Attended Episodes of Respiratory Syncytial Virus-related Lower Respiratory Tract Infection Among Infants in the United StatesPLOS ONE

Dear Dr. Gantenberg,

Thank you for submitting your manuscript to PLOS ONE. After careful consideration, we feel that it has merit but does not fully meet PLOS ONE’s publication criteria as it currently stands. Therefore, we invite you to submit a revised version of the manuscript that addresses the points raised during the review process.

We look forward to receiving your revised manuscript.

Kind regards,

Farhana Haque, MBBS MPH MSc PhD

Academic Editor

PLOS ONE

“This work was supported by Sanofi and AstraZeneca.”

Reviewers' comments:

Reviewer's Responses to Questions

**Comments to the Author**

1. Is the manuscript technically sound, and do the data support the conclusions?

Reviewer #1: Partly

Reviewer #2: Yes

2. Has the statistical analysis been performed appropriately and rigorously? 

Reviewer #1: N/A

Reviewer #2: Yes

3. Have the authors made all data underlying the findings in their manuscript fully available?

Reviewer #1: Yes

Reviewer #2: No

4. Is the manuscript presented in an intelligible fashion and written in standard English?

Reviewer #1: Yes

Reviewer #2: Yes

5. Review Comments to the Author

Reviewer #1: This study finds a new information that not only high risk babies but healthy term infants and infants with less severe comorbidities account for the majority of healthcare visits during acute MA RSV LRTI episodes.

Hence I have few observation

Abstract

Conclusion could include difference between payers

Methods:

Sample size calculation need to mention.

Statistical tools used in this study is missing

Reviewer #2: The study is an important addition to the RSV literature, documenting the healthcare burden among infants' first RSV LRTI episode across commercial and Medicaid data sources. I appreciate the authors' discussion of the study limitations, which are extensive and common in studies of claims data, but not often described so thoroughly. This study makes use of proprietary data sources from Merativ and Optum, and therefore, the data cannot be made publicly available.

6. PLOS authors have the option to publish the peer review history of their article (what does this mean?). If published, this will include your full peer review and any attached files.

Reviewer #1: No

Reviewer #2: No

---

## [Author Response · Author response to Decision Letter 0]

25 Sep 2024

We appreciate the reviewers' favorable disposition toward our original submission, as well as their suggestions for improvement. We hope our responses and revisions address the comments we received. In addition, we have made a minor correction to the table in the supplement to remove an ICD code that was double-listed (see tracked version).

RESPONSES TO REVIEWERS

Journal requirements

1. Please ensure that your manuscript meets PLOS ONE’s style

 requirements, including those for file naming. The PLOS ONE style

 templates can be found at

 and

 AUTHOR REPLY:

 We have reviewed the PLOS ONE style guides again and updated

 formatting and file-naming as needed.

 “This work was supported by Sanofi and AstraZeneca.”

 Please state what role the funders took in the study. If the funders

 had no role, please state: “The funders had no role in study design,

 data collection and analysis, decision to publish, or preparation of

 the manuscript.”

 Please include this amended Role of Funder statement in your cover

 letter; we will change the online submission form on your behalf.

 AUTHOR REPLY:

 We have updated our funding statement to include the additional

 information requested. The statement now reads as follows:

 This work was supported by Sanofi and AstraZeneca. Employees of

 Sanofi (RvA, CBN) contributed to the study design, interpretation

 of results, and manuscript review. Sanofi also provided Brown

 researchers access to the Optum Clinformatics Data Mart. Both

 Sanofi and AstraZeneca conducted intellectual property compliance

 review prior to submission. The decision to submit for publication

 rested solely with the Brown research team.

3. We note that you have indicated that there are restrictions to data

 sharing for this study. For studies involving human research

 participant data or other sensitive data, we encourage authors to

 share de-identified or anonymized data. However, when data cannot be

 publicly shared for ethical reasons, we allow authors to make their

 data sets available upon request. For information on unacceptable

 data access restrictions, please see

http://journals.plos.org/plosone/s/data-availability#loc-unacceptable-data-access-restrictions.

 Before we proceed with your manuscript, please address the following

 prompts:

 1. If there are ethical or legal restrictions on sharing a

 de-identified data set, please explain them in detail (e.g.,

 data contain potentially identifying or sensitive patient

 information, data are owned by a third-party organization, etc.)

 and who has imposed them (e.g., a Research Ethics Committee or

 Institutional Review Board, etc.). Please also provide contact

 information for a data access committee, ethics committee, or

 other institutional body to which data requests may be sent.

 2. If there are no restrictions, please upload the minimal

 anonymized data set necessary to replicate your study findings

 to a stable, public repository and provide us with the relevant

 URLs, DOIs, or accession numbers. Please see

http://www.bmj.com/content/340/bmj.c181.long for guidelines on

 how to de-identify and prepare clinical data for publication.

 For a list of recommended repositories, please see

https://journals.plos.org/plosone/s/recommended-repositories.

 You also have the option of uploading the data as Supporting

 Information files, but we would recommend depositing data

 directly to a data repository if possible.

 Please update your Data Availability statement in the submission

 form accordingly.

 AUTHOR REPLY:

 We have updated our Data Availability Statement in to provide

 additional information on accessing the MarketScan and Optum

 databases. Our updated statement reads as follows:

 Our funding agreements with Sanofi and AstraZeneca did not impose

 any restrictions on data sharing. However, we are unable to share

 insurance claims data because these data are owned by third

 parties and governed by the data owners’ usage restrictions.

 Researchers should contact Merative or Optum to purchase access to

 the claims data sets used in this paper. The surveillance data on

 respiratory syncytial virus we used to determine RSV season start

 and end dates by geographic region can be requested from the

 Centers for Disease Control and Prevention. We have described the

 method to calculate these dates in a prior publication referenced

 in the Methods. S5 Code contains code and select (public) data

 used to generate the CDM analytic dataset and results (code for

 the Merative analyses cannot be shared due to their licensing

 agreements with SAS), as well as the scripts and summary data

 files that produce the manuscript, including tables and figures.

4. Please include captions for your Supporting Information files at the

 end of your manuscript, and update any in-text citations to match

 accordingly. Please see our Supporting Information guidelines for

 more information:

http://journals.plos.org/plosone/s/supporting-information.

 AUTHOR REPLY:

 We have added the Supporting Information captions to the end of the

 main manuscript, as requested.

5. Please review your reference list to ensure that it is complete and

 correct. If you have cited papers that have been retracted, please

 include the rationale for doing so in the manuscript text, or remove

 these references and replace them with relevant current references.

 Any changes to the reference list should be mentioned in the

 rebuttal letter that accompanies your revised manuscript. If you

 need to cite a retracted article, indicate the article’s retracted

 status in the References list and also include a citation and full

 reference for the retraction notice.

 AUTHOR REPLY:

 Thank you. We have reviewed our reference list as requested and did

 not identify any necessary changes.

# Reviewers’ Comments

## Reviewer 1

This study finds a new information that not only high risk babies but

healthy term infants and infants with less severe comorbidities account

for the majority of healthcare visits during acute MA RSV LRTI episodes.

Hence I have few observations

1. Abstract: Conclusion could include difference between payers

AUTHOR REPLY

We appreciate the suggestion and have added the following passage to our

abstract’s conclusion:

 The percentage of MA RSV LRTI episodes involving at least 2 visits to

 a healthcare setting may vary by insurance claims database, even

 between commercial payers.

1. Methods

 1. Sample size calculation need to mention.

 AUTHOR REPLY

 We appreciate the suggestion. Sample size calculations are typically

 used in hypothesis testing to determine the sample size necessary to

 achieve a prespecified statistical power to detect a given effect

 size (/i.e., a difference in outcomes between two groups). Our goals

 in this descriptive study involved neither hypothesis testing nor

 causal inference, and therefore, we believe sample size calculations

 do not apply.

 1. Statistical tools used in this study is missing

 AUTHOR REPLY 

 Given our descriptive aims, we limited our statistical analysis to

 calculating 95% confidence intervals around estimates of

 proportions, in order to quantify variability due to random error.

 The section entitled Descriptive Analysis already provides a

 reference for the approximate intervals we used, and we are unsure

 what to add, as these were the only statistical tools (besides

 tabulation) used in the study. We did not conduct statistical tests

 because these did not apply given the aims of the study.

## Reviewer 2

The study is an important addition to the RSV literature, documenting

the healthcare burden among infants’ first RSV LRTI episode across

commercial and Medicaid data sources. I appreciate the authors’

discussion of the study limitations, which are extensive and common in

studies of claims data, but not often described so thoroughly. This

study makes use of proprietary data sources from Merativ and Optum, and

therefore, the data cannot be made publicly available.

AUTHOR REPLY

We appreciate the reviewer’s positive assessment of our study and are

happy that they found our discussion of the limitations of claims data

to be helpful.

---

## [Editor Report · Decision Letter 1]

28 Oct 2024

Healthcare Utilization During Acute Medically Attended Episodes of Respiratory Syncytial Virus-related Lower Respiratory Tract Infection Among Infants in the United States

PONE-D-24-01606R1

Dear Dr. Gantenberg,

We’re pleased to inform you that your manuscript has been judged scientifically suitable for publication and will be formally accepted for publication once it meets all outstanding technical requirements.

Kind regards,

Farhana Haque, MBBS MPH MSc PhD

Academic Editor

PLOS ONE
---

## [Editor Report · Acceptance letter]

23 Jan 2025

PONE-D-24-01606R1 

PLOS ONE

Dear Dr. Gantenberg, 

I'm pleased to inform you that your manuscript has been deemed suitable for publication in PLOS ONE. Congratulations! Your manuscript is now being handed over to our production team.

Kind regards, 

on behalf of

Dr Farhana Haque 

Academic Editor

PLOS ONE